# Process Evaluation of the Metal-Organic Frameworks for the Application of Personal Protective Equipment with Filtration Function

**DOI:** 10.3390/polym10121386

**Published:** 2018-12-14

**Authors:** Yan Hong, Chunyu Liu, Xuechun Cao, Yu Chen, Chen Chen, Yan Chen, Zhijuan Pan

**Affiliations:** 1College of Textile and Clothing Engineering, Soochow University, Suzhou 215021, China; yannichonghk@gmail.com (Y.H.); xuechuncao@suda.edu.cn (X.C.); yanchen@suda.edu.cn (Y.C.); 2College of Chemistry, Chemical Engineering and Materials Science, Soochow University, Suzhou 215021, China; chunyuliu94@163.com; 3School of Fashion Engineering, Shanghai University of Engineering Science, Shanghai 201600, China; yuchenbonjour@hotmail.com; 4GEMTEX, ENSAIT, 2 allée Louise et Victor Champier, 59056 Roubaix CEDEX 1, France; chen.chen@ensait.fr

**Keywords:** MOFs, electrospinning, hot-pressing, PM2.5, PM10, PM1.0

## Abstract

Metal-organic frameworks (MOFs) have been regarded as an ideal material for the development of functional textiles with filtration function. Such functional textiles with filtration function can be further used to develop personal protective equipment, such as protective masks. This paper focuses on the comparisons of different processes when applying MOFs to conventional textiles. Two different processes existing in the literature, namely the electrospinning method and hot-pressing method, are discussed in this paper. Materials loaded with MOFs developed with these two processes are evaluated and compared, regarding the adsorption of dyes in water and the removal of pollutants. Experiment results indicate that the hot-pressing method is more advantageous when applying MOF to textiles, in terms of adsorption and removal efficiency.

## 1. Introduction

With the rapid development of global industries, personal protective equipment has been increasingly required to avoid environmental problems, both in the industrial workplace and daily life [1]. Air pollution, as one of the major pollutions created by industrial processes, has been increasingly raising attention [2]. As a major air pollution source, particulate matters (PMs) have become one of the most serve environmental issues, especially in China [3]. PM2.5 and PM10, particulate matter (PM) with an aerodynamic diameter less than 2.5 μm and 10 μm, cause serious harm to human health [4]. Various diseases, such as cardiovascular and respiratory diseases, increasing morbidity, and even mortality, can be caused if the human body is exposed to PM2.5 for long-term [5].

With this in mind, personal protective equipment with filtration function, which is able to avoid air pollution, has been developed. The referenced personal protective equipment includes protective masks [6] and protective garments [7]. For example, Steve Zhou developed a respiratory face mask. Such a mask is able to capture air pollutants and pathogens, including human influenza and rhinoviruses [6]. For this kind of personal protective equipment, the most complicated task is the design for the filtration of PM [8]. The traditional filtration materials are developed using textile techniques [9]. Textiles with higher density have been chosen to realize the filtration function, which filters air pollution through the voids of the fabric [9]. However, such material not only has very poor ventilation, but also greatly affects thermal comfort of the wearer, especially for protective garments.

As novel crystalline porous materials, metal-organic frameworks (MOFs), have drawn wide attention in the past decades [10,11,12,13]. Metal-organic frameworks (MOFs) are synthesized from metal ions or clusters and organic links [14]. The organic ligands in MOFs are able to form highly regular networks, which are realized by the connection to metal ions [15]. Metal-organic frameworks (MOFs) have several distinct features, such as tunable pore functionality, highly diversified structures, high surface area, and high degree of crystallinity. These features ensure MOFs have the ability of sensors, gas storage, catalysis, gas separation, and drug delivery [9]. There are different researchers using MOFs as a new solution to replace traditional textiles as filtration materials for personal equipment [16]. For example, Chang used metal-organic frameworks (MOFs) to develop a highly adaptive organic-inorganic hybrid fibrous substrate through an electrospinning process [16]. Such MOF materials have been further used to develop facial masks, which can protect against PM2.5.

In general, based on existing literature, there are two different processes when applying MOFs to textiles. The first process is that MOF materials are mixed with a conventional electrospinning solution [16]. Electrospinning uses electric force to draw charged threads of polymer solutions or polymer-melts up to fiber diameters [17]. It is a common fiber production method. Electrospinning shares characteristics of both electro-spraying and conventional solution dry spinning of fibers. Through an electrospinning process, textiles with filtration functions are developed [18]. The second process is based on the thermal modification of traditional textiles [19]. MOFs will be attached to the textile surface directly through weak hydrogen bonding. The filtration properties of MOFs will be offered to the textiles correspondingly. However, there is no existing research focusing on the comparison of these two different processes when applying MOFs to conventional textiles.

In this research, in order to widely enhance the application of MOFs on conventional textiles, we evaluated the two different processes. This paper is organized as follows: Section 2 describes the comparison experiment, the involved materials, and methods. Section 3 presents the experiment results and related discussion. Section 4 concludes this paper.

## 2. Materials and Methods 

### 2.1. Materials 

All involved chemicals and solvents were obtained from commercial suppliers including Alfa Aesar (Heysham, Lancashire, UK), Sigma-Aldrich (Shanghai, China), and Beijing Chemical Reagent Company (Beijing, China). The involved chemicals and solvents obtained from these suppliers were directly applied to the experiments and used without further purification.

Zn(NO_3_)_2_·4H_2_O (99.99%), 2-methylimidazole (98%), and polyacrylonitrile (PAN) polyethylene glycol (*M_W_* = 150,000 g/mol) were purchased from Aladdin Industrial Corporation (Shanghai, China). Deionized water was used throughout all the experiments. Non-woven fabric, which is made of polyethylene terephthalate, was purchased from a commercial source. Non-woven fabric substrates were washed with ethanol and then dried at 60 °C for 3 h before being used.

### 2.2. Methods

The PANalytical X’Pert PRO MPD system (PW3040/60, Empyream, Netherlands) with Cu-Kα radiation was used to realize the Powder X-ray diffraction (PXRD) measurements. UV−Vis spectra were measured on a Varian Cary-50 UV−Vis spectrophotometer (Los Angeles, CA, USA). The scanning electron microscopy (SEM) measurement was performed using a JSM-5600LV instrument (Chiyoda-ku, Tokyo, Japan). 

### 2.3. Synthesis of ZIF-8

First, 1.291 g of Zn(NO_3_)_2_·4H_2_O was dissolved in 100 mL of methanol, and 1.621 g of 2-methylimidazole was dissolved in 100 mL of methanol. Then, the two solutions were mixed by stirring and kept statically at room temperature for a period of 24 h. The resultant powders were collected by centrifugation and thoroughly washed with methanol. The product was dried at 100 °C for a period of 12 h in a vacuum oven.

### 2.4. Preparation of Experiment Materials

The ZIF-8/PAN solution was as follows: 120 mg of PAN (*M_W_* = 150,000 g/mol) and 40 mg ZIF-8 was added in 1 mL of DMF. The mixture was stirred at 60 °C for a period of 20 min to form a homogeneous solution with a ZIF-8 loading of 20 wt %. The ZIF-8/PAN solutions of different concentrations (40 wt % and 60 wt %) were prepared in the same way.

The electrospinning solution was electrospun onto non-woven fabric substrates at a voltage of 18 kV. The flow rate was 1 mL/h. The spinneret diameter was 0.6 mm. The distance between the spinneret and the collector was set at 15 cm. The involved non-woven fabric substrates were prepared to the size of 20 × 20 cm. ZIF-8-E1 to E3, which corresponding to ZIF-8 loading of 20 wt %, 40 wt %, and 60 wt %, were afforded after electrospinning for 1 h. 

Finally, 40 mg of ZIF-8 was added into 10 mL C_2_H_5_OH, and the mixture was stirred for a period of 2 h at room temperature. The non-woven fabric substrates were immersed in the mixture for 1 h. Next, the section was covered with a piece of aluminum foil, after which they were packed and heated with an electric iron at 100 °C for 10 min. This sample was designated as ZIF-8-P1. Samples ZIF-8-P2 and ZIF-8-P3 were then realized by repeating the procedure either once or twice.

### 2.5. Air Purification Performance Evaluation

In order to assess the air purification performance, a home-used air purifier (EC66, Yichuang Lifestyle Co. Ltd., Beijing, China) with a fibrous membrane prepared with our materials was used. A polluted environment was created with normal cigarettes in a confined space with a volume of 0.125 m^3^. Then the vehicle filter was placed in the polluted environment. The PM2.5 concentration was detected by an air quality detector (MEF-550, Sensology Institute, Michigan, MI, USA). The concentration of PM2.5 was detected at different times to record the air purification performance of the fibrous membranes.

### 2.6. Dye Adsorption on ZIF-8-E3 and ZIF-8-P3

The adsorption behaviors of methylene blue from aqueous solutions on ZIF-8-Filbers were studied at room temperature. In the experiment process, composite sorbent was added into 100 mL of 100 mg L^−1^ dye solution. The solutions were oscillated until equilibrium was reached. Next, the remaining dye solution was collected and analyzed. A UV-240 UV-vis spectrometer was utilized to record the absorbance change of the dye solution. 

The dye isothermal adsorption for ZIF-8 composite was measured by varying the initial dye concentrations. In detail, ZIF-8-Filbers were added into 100 mL of aqueous solutions with different dye concentrations (ranging from 10 to 1000 mg L^−1^) at room temperature. 

The dynamic property of adsorption was also analyzed. The mixtures with the ZIF-8 composite and dyes were continuously oscillated for different periods, in order to investigate the effect of contact time on adsorption. Following the method described above, the concentrations of the remaining dyes in the aqueous solution were measured. The ZIF-8 composites were immersed in EtOH for 24 h under oscillation. There were washing processes with DI water for regeneration, which were repeated several times. After the regeneration process, a drying process for the sorbent was performed in an oven at 80 °C.

## 3. Results and Discussion

### 3.1. Discuss on the Synthesis of Materials

As is reported in the literature, electrospinning is able to be feasibly create continuous nanofibers containing MOF nanoparticles. Such a method is demonstrated to be one of the simplest top-down methods that ensures an easy generation of nanofibers from a wide variety of materials. Polymers developed by this method were proven to be efficient for many applications, such as filtration or controlled drug-release. In this research, we also presented ZIF-8 fibers via electrospinning through an electrospinning process. Firstly, ZIF-8 were synthesized according to typical methods [14]. Next, the prepared ZIF-8 fibers were dispersed in the PAN solutions for electrospinning to afford ZIF-8-E1 to E3, which corresponded to ZIF-8 loading of 20 wt %, 40 wt %, and 60 wt %. PAN was chosen to be the matrix for its excellent filtering ability for PM capture [14]. Figure 1a presents the general principle of the involved electrospinning method.

Even though it is easy to obtain large-area non-woven fibrous mats through electrospinning, which could facilitate many applications, there are still some obstacles since it is a one-step electrospinning method. Using this method, most MOF particles are encapsulated in the fibers and are unavailable for PM2.5 adsorption. Only at high loadings can there be enough MOF particles exposed to the fiber surface so that the material can contact PM2.5, thus resulting in a high surface area for practical use. 

In this research, we also introduced the hot-pressing (HoP) method to provide a comparison with other composition methods. Under a certain temperature and pressure, MOF was reported to react with the surface functional groups or metal sites on the surface of the substrates to form MOF fibers. Consequently, this HoP method can provide high affinity for MOF nanocrystals to attach onto the surface of flexible substrates. Subsequently, the resilience of such MOF-coated devices can be enhanced. Figure 1b presents the working principle of the hot-pressing process. The advantage of this method is that it can load the material onto the surface to the greatest extent. Using this method, more active sites can be exposed on the surface of the desired material.

### 3.2. Discussion on the Structure

From Figure 2, it can be seen that these electrospun MOF fibers present a well-defined morphology. The diameter of the obtained textiles is about 200–300 nm, and it can be adjusted by changing the experimental conditions (such as changing voltage and concentration of the spinning solution). Under high load conditions, the surface particles of the obtained material can be evenly distributed. The particles did not agglomerate under high load conditions, which proved ZIF-8 has good dispersion and compatibility in textiles. 

From Figure 3, it can be found that using hot-pressing (HoP) method, ZIF-8 forms relatively uniform particles on the surface of the textile. The diameter of these particles is about 100 nm. With an increase in the number of cycles, the surface particles become denser, while the particles are also evenly distributed without agglomeration.

Through the comparison between XRD patterns of the ZIF-8 nanofibers and the pure nanoparticles, it can be concluded that the crystal structure was preserved.

Next, we carried out a BET test. The test results are shown in Figure 4. It can be seen that the specific surface area of ZIF-8 powder is 1036 m^2^/g, while the specific surface areas of ZIF-8-E3 and ZIF-8-P3 are 467 m^2^/g and 716 m^2^/g, respectively.

### 3.3. Adsorption Properties of ZIF-8 Composite for Dyes

In order to test the water adsorption properties of the composite, we chose ZIF-8-E3 and ZIF-8-P3 to test its adsorption properties for methyl blue. The saturation adsorption was performed with a 100 mg L^−1^ MB solution. UV/Vis spectroscopy was utilized to monitor the concentration changes of MB during the adsorption procedure. 

The adsorption process was completed within just over 2 h by using ZIF-8-E3 and ZIF-8-P3 as the sorbents (see in Figure 5). From Figure 5, we can see that the decrease of MB was dramatic, and the removal rate reached 96.3% for ZIF-8-E3 and 98.5% for ZIF-8-P3. For ZIF-8 powder, removal of 89.7% was obtained after 2.5 h. The fast adsorption rate within short time of ZIF-8-E3 and ZIF-8-P3 was due to the fact that the nanocrystals of ZIF-8 were more dispersed and subsequently may speed up the mass transfer to obtain rapid adsorption. By contrast, the aggregated particles in the sample ZIF-8 powder may result in lower adsorption rate. Further, all adsorptions belong to the first order reaction.

### 3.4. Air Purification Performance

The air purification performance evaluation was performed in a real polluted environment. A filter element for a car air purifier was prepared using the membrane (the apparatus is displayed in Figure 6) in the evaluation experiment. A polluted environment was created by burning cigarettes in a confined space with a volume of 0.125 m^3^. The PM2.5 concentration was initially set to over 999 g/m^3^.

Next, ZIF-8-E3 and ZIF-8-P3 were employed to remove particulate matter in the simulated environments. Both ZIF-8-E3 and ZIF-8-P3 showed great performances, which suggests the higher efficiencies stem from the unique properties endowed by ZIF-8. 

Figure 7 presents PM filtration efficiencies of ZIF-8-P3 and ZIF-8-E3. From the curve of PM2.5 over time, we found that ZIF-8-E3 intercepted PM10 faster, while the interception of PM2.5 and PM1.0 was relatively slow. The phenomenon is similar to PAN, which is reported in the literature. However, the effect was better than the PAN because it is not loaded. From this phenomenon, we hypothesize that in this process, the filtration of PM10 caused by the pores of the fiber occurred first, followed by the adsorption of ZIF-8 to PM2.5 and PM1.0. At 30 min, the removal effect of PM10 was approximately 88.64%, while the removal of PM2.5 and PM1.0 was approximately 78.35% and 78.26%, respectively.

When using P3 materials, the situation was significantly different. The filtration rate for small particle contaminants was significantly greater than for large particles. This is due to the fact that the pores of the substrate are too large, resulting in a weaker interception effect on large particles than E3. The situation for small particles is the opposite, since the ZIF-8 material loaded in the hot-pressing (HoP) method is mostly distributed on the surface of the material, which increases the specific surface area. This condition enhances the filtration performance for small particles. At 30 min, the removal effect of PM10 was about 86.52%, while the removal of PM2.5 and PM1.0 was about 96.24% and 94.78%, respectively.

## 4. Conclusions

In this paper, two different methods, namely electrospinning and hot-pressing (HoP), were used to synthesize ZIF-8-loaded textile materials. The performance of the materials, obtained from the two methods, regarding the adsorption of dyes in water and the removal of pollutants in air, were compared. From our research, it was found that the hot-pressing method was more advantageous when applying MOF to textiles, in terms of adsorption and removal efficiency. Textiles loaded with MOF have improved adsorption and filtration performance. Such materials can be further used in personal protective equipment development. It also provides reference for the development and practical application of composite materials.

## Figures and Tables

**Figure 1 polymers-10-01386-f001:**
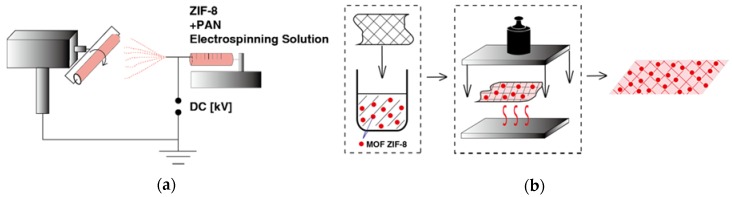
(**a**) An example of the spinning process; (**b**) an example of the hot-pressing process.

**Figure 2 polymers-10-01386-f002:**
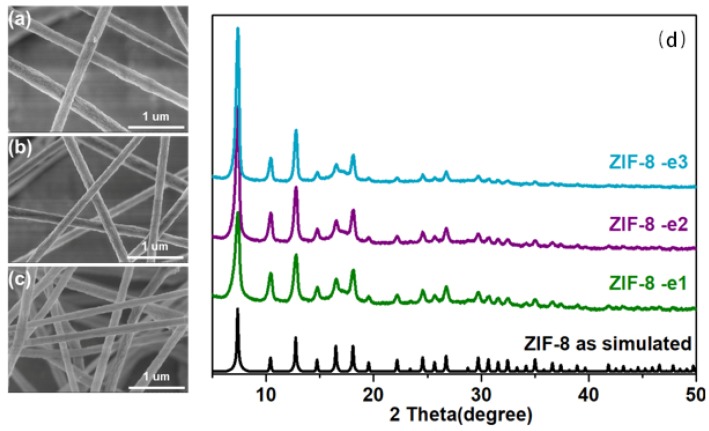
Characterizations of ZIF-8-E1 (**a**); ZIF-8-E2 (**b**) and ZIF-8-E3 (**c**); (**d**) PXRD patterns of ZIF-8-E1 to E3.

**Figure 3 polymers-10-01386-f003:**
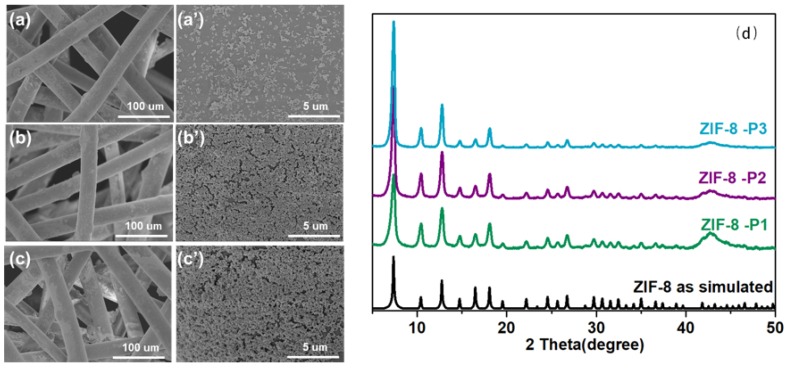
Characterizations of ZIF-8-P1 (**a**; **a**′), ZIF-8-P2 (**b**; **b**′), and ZIF-8-P3 (**c**; **c**′); (**d**) PXRD patterns of ZIF-8-P1 to P3.

**Figure 4 polymers-10-01386-f004:**
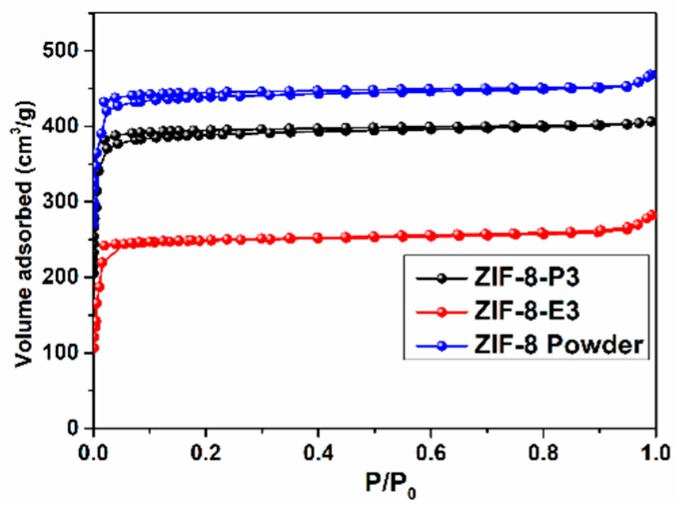
N_2_ adsorption−desorption analysis of ZIF-8 powder, ZIF-8-P3, and ZIF-8-E3.

**Figure 5 polymers-10-01386-f005:**
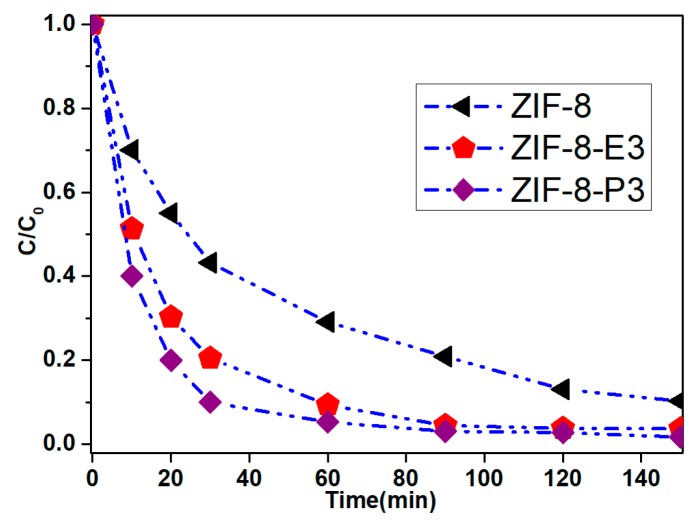
Absorption properties of ZIF-8; ZIF-8-E3 and ZIF-8-P3 in the absorption of MB in water.

**Figure 6 polymers-10-01386-f006:**
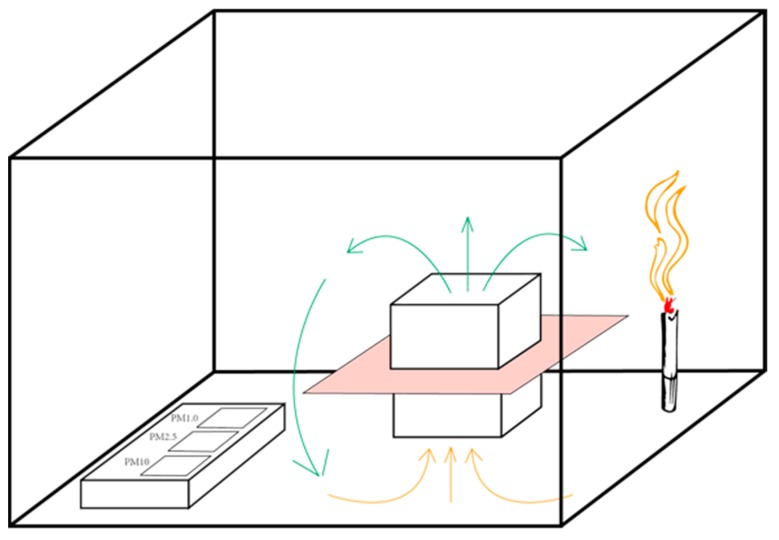
The apparatus of the air purification preformation evaluation.

**Figure 7 polymers-10-01386-f007:**
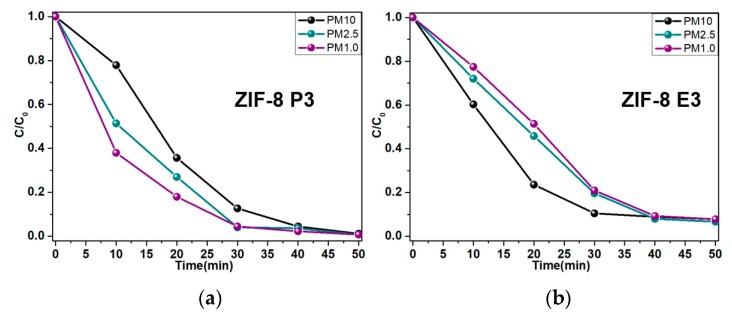
PM filtration efficiencies of ZIF-8-P3 (**a**) and ZIF-8-E3 (**b**).

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
