# Peer review of "Process Evaluation of the Metal-Organic Frameworks for the Application of Personal Protective Equipment with Filtration Function"

_polymers, 2018, doi:10.3390/polym10121386_

Round 1
Reviewer 1 Report
Hong et al. present a comparison of electrospinning and hot-pressing (HoP) to synthesize ZIF-8-loaded textiles, capable of environmental applications. The work is written very fluently and almost publishable as it is. Below some minor comments/suggestions:
The introduction is concise and well-written but I recommend to also cover the principle of electrospinning. Relevance reference are: Journal of Fluid Mechanics 2004, 516;349; ACS Appl. Mater. Inter. 2017, 9, 24100; Adv. Funct. Mater. 2018, 44, 1804138.
General comment 2.4 and 2.5: the titles are probably for a quick reader somewhat confusing
L 76 g/mol is lacking; also further
How relevant is the spinning in DMF in view of the environmental context?
L 96: please check “The electrospinning solutions solution was electrospun”
A somewhat more detailed discussion of Figure 2-3 would be beneficial.
Figure 4: perhaps report firs order rate coefficients related to the efficiency
Author Response
Hong et al. present a comparison of electrospinning and hot-pressing (HoP) to synthesize ZIF-8-loaded textiles, capable of environmental applications. The work is written very fluently and almost publishable as it is.
Response:
Thanks for your kind revise.
The introduction is concise and well-written but I recommend to also cover the principle of electrospinning. Relevance reference are: Journal of Fluid Mechanics 2004, 516;349; ACS Appl. Mater. Inter. 2017, 9, 24100; Adv. Funct. Mater. 2018, 44, 1804138.
Response:
Thank you for your suggestions. Suggested references have been added. The following sentences have been added:
Electrospinning uses electric force to draw charged threads of polymer solutions or polymer-melts up to fiber diameters [15]. It is a common fiber production method. Electrospinning shares characteristics of both electro-spraying and conventional solution dry spinning of fibers.
General comment 2.4 and 2.5: the titles are probably for a quick reader somewhat confusing
Response:
The title of 2.4 has been modified into: “Preparation of experiment materials.”
The title of 2.5 has been deleted.
L 76 g/mol is lacking; also further
Response:
Related content has been modified.
How relevant is the spinning in DMF in view of the environmental context?
Response:
As discussed in Section 3.1, PAN has reported to be environmentally friendly based on literature. It has the function of intercepting PM2.5 as well.
PAN has good solubility in DMF. That is why we choose DMF.
DMF is completely soluble in water. In industrial process, through a water wash process, DMF can be completely removed.
That is why DMF is chosen.
L 96: please check “The electrospinning solutions solution was electrospun”
Response:
Related content has been modified.
A somewhat more detailed discussion of Figure 2-3 would be beneficial.
Response:
More information has been given. Please check the manuscript.
Figure 4: perhaps report firs order rate coefficients related to the efficiency
Response:
All adsorptions belong to the first order reaction.
Reviewer 2 Report
This is an intersting work which describes the Metal−Organic Frameworks for the Application to Personal Protective Equipment with Filtration Function. The results are intersting. We think it can be published after minor revision.
1. It will be better to add the bet data here.
2. some recent achivement of MOF can be cited in the introduction part :Applied Catalysis B: Environmental 240 (2019) 92–101; J. Mater. Chem. A, 2018,6, 20304-20312. Journal of Catalysis 361 (2018) 238–247;
Author Response
This is an interesting work which describes the Metal−Organic Frameworks for the Application to Personal Protective Equipment with Filtration Function. The results are interesting. We think it can be published after minor revision.
Response:
Thanks for your time and kind acceptance.
It will be better to add the bet data here.
Response:
It has been added. Please check in the manuscript.
2. some recent achivement of MOF can be cited in the introduction part: Applied Catalysis B: Environmental 240 (2019) 92–101; J. Mater. Chem. A, 2018,6, 20304-20312. Journal of Catalysis 361 (2018) 238–247;
Response:
Thank you for your suggestions. Suggested references have been added.